# Preimplantation Testing of Human Blastomeres for Aneuploidy Increases IVF Success in Couples Where Male Partners Had Abnormal Semen Parameters

**DOI:** 10.3390/biomedicines13051191

**Published:** 2025-05-13

**Authors:** Mahira Ismayilova, Aytakin Hasanova, Andrei Semikhodskii

**Affiliations:** 1Central Clinic Hospital, 76 Parliament Ave., Baku AZ1006, Azerbaijan; mahiremk@hotmail.com; 2Department of Medical Biology and Genetics, Azerbaijan Medical University, Baku AZ1001, Azerbaijan; aytakin_hasanova@mail.ru; 3Medical Genomics Ltd., 134 Somerset Road, London SW19 5HP, UK

**Keywords:** asthenozoospermia, male infertility, PGT-A, oligospermia, oligozoospermia, teratozoospermia

## Abstract

**Background/Objectives:** Male infertility is becoming a serious problem affecting about 7% of all men worldwide and is a major or contributory factor in 50% of infertile couples overall. Men with abnormal semen parameters have a significantly increased risk of aneuploidy, presenting a serious concern in programmes of assisted reproductive technologies. Recently, the introduction of preimplantation genetic testing for aneuploidies (PGT-A) has increased the pregnancy rate and live births. We investigated the effect of PGT-A on the success of IVF treatment in couples with the male factor of infertility. **Methods:** Two experimental groups and one control group were studied: Group A (110 couples)—male partners with abnormal semen parameters, with PGT-A; Group B (110 couples)—male partners with abnormal semen parameters, without PGT-A; and Group C (105 couples)—control, male partners with normal spermograms, with PGT-A. A Day 3 blastomere biopsy was followed by FISH-based PGT-A. A total of 880 embryos from Group A and 890 embryos from Group C was analysed. **Results:** In patients with abnormal semen parameters, embryonic aneuploidy was twice as common compared to the control (13.6% vs. 5.8%, *p* < 0.001). Group B had the lowest clinical pregnancy rate (28.2%), with two out of three pregnancies ending in a miscarriage. Only 10% of IVF cycles in this group resulted in live birth compared with 35.5% for Group A and 49.5% for Group C. **Conclusions:** Our data demonstrate that PGT-A screening as part of IVF treatment drastically increases the clinical pregnancy rate and chances of live birth in couples where male partners have semen abnormality.

## 1. Introduction

Male infertility is becoming a common problem. It is estimated that 7% of men have fertility problems [1]. According to a recent study, a male factor is solely responsible for about 20% of infertility in couples [2] and is a major or contributory factor in 50% of infertile couples overall [3]. Considering the etiological and pathogenetic aspects of male infertility, not only quantitative changes in the number and degree of sperm motility but also morphological, genetic, and functional disorders must be taken into account [4]. In men with abnormal semen parameters (e.g., oligospermia, asthenospermia, etc.), the risk of aneuploidy is significantly increased [5], presenting a serious concern in programmes of assisted reproductive technologies (ART). Fertilisation and cleavage rates, the quality of embryos, and blastocyst development rates are significantly reduced as semen quality decreases [6]. Recently, the widespread introduction of preimplantation genetic testing for aneuploidies (PGT-A) into ART treatment has made it possible to select euploid embryos for transfer, thus leading to an increase in the pregnancy rate and live births (LBs) [7,8]. At the same time, there are still open questions regarding some etiological aspects of male infertility, as well as the frequency and types of aneuploidies in embryos obtained during in vitro fertilisation (IVF) treatment in couples where the male partner had abnormal semen parameters [9].

Over the past three decades, the methodologies used to analyse human embryos for the presence of aneuploidies have undergone revolutionary changes. Some of the first approaches were based on conventional karyotyping, which required cells at the metaphase division stage. This method had serious disadvantages—only 24–36% of karyotyped embryos produced metaphases of sufficient quality for accurate chromosome analysis [10]. It was possible to analyse only developing cells, and because of difficulties in obtaining an optimal distribution of chromosomes, there were problems in identifying individual chromosomes [11]. In 1997, Gianaroli and colleagues [12] proposed preimplantation genetic testing of embryos using fluorescent in situ hybridisation (FISH) on trophectodermal cells collected at the cleavage stage. This approach allowed obtaining chromosomal information independent of the state of development of individual cells, allowing analysis of chromosome numbers in both metaphase and interphase nuclei. We used FISH analysis to detect embryonic aneuploidies in the current work.

The objective of the present study was to evaluate the efficiency of PGT-A in couples where the male partner has oligozoospermia, asthenozoospermia, or other abnormal semen parameters. To avoid freezing embryos, a Day 3 blastomere biopsy was performed, coupled with PGT-A analysis using FISH, followed by a Day 5 embryo transfer. The results show the outcome of including PGT-A in the IVF protocol when treating patients with the male factor of infertility in a private clinical environment and will be beneficial as a guide to specialists on the strategy of ART treatment as well as for counselling these patients.

## 2. Materials and Methods

All research was performed in accordance with relevant guidelines and regulations.

### 2.1. Participants

For the study, 325 married couples who presented themselves for IVF treatment at the Central Clinic Hospital, Baku, Azerbaijan, were selected and allocated into two experimental groups (Group A and Group B) and one control (Group C) group. Group A consisted of 110 couples where the male partners had abnormal semen parameters, who underwent IVF coupled with PGT-A. Group B included 110 couples where the male partners had abnormal semen parameters, who underwent IVF without PGT-A (the patients did not consent to PGT-A). Group C consisted of 105 couples where the male partner had normal spermograms, who underwent IVF coupled with PGT-A. Female partners from Group A and Group B were reproductively healthy. Female partners from Group C had the tubal–peritoneal factor of infertility due to past ectopic pregnancy, pelvic inflammatory disease, or other conditions affecting the patency of the fallopian tubes. We used criteria for male abnormal sperm parameters recommended by the World Health Organisation (WHO) [13].

Before proceeding with IVF, all female patients were investigated for follicle-stimulation hormone (FSH), luteinising hormone (LH), estradiol (E2), thyroid-stimulating hormone (TSH), free triiodothyronine (fT3), free thyroxine (fT4), prolactin, anti-Müllerian hormone (AMH), testosterone, anti-thyroid peroxidase (Anti-TPO) antibodies, toxoplasmosis, other (syphilis, varicella zoster, parvovirus B19), rubella, cytomegalovirus (CMV), and herpes (TORCH) infections and underwent a Papanicolaou (Pap) test, karyotype evaluation, and measurement of blood vitamin D levels. The reproductive organs were physically examined using hysterosalpingography and hysteroscopy with pathohistological examination of the endometrial biopsy sample. In men, a sperm count was carried out, and sperm morphology was studied. In addition, the karyotype of the male partner was investigated. Both male and female participants were investigated for sexually transmitted infections.

The demographics of the study groups are given in Table 1. The age of the female partners included in the study was 21–43 years; the age of the male partners was 27–52 years. There were no statistically significant differences between the groups with regard to the age of the male and female partners as determined by the two-sample K-S test.

Couples where the woman had ovarian dysfunction or one of the partners had thyroid dysfunction, diabetes mellitus, autoimmune diseases, or cancer, was a known carrier of a monogenic disease, or had an abnormal karyotype were excluded from the study. To avoid bias, couples in which one of the spouses smoked or had alcohol or drug addiction were also excluded from the study group as smoking, alcohol, and substance dependencies are known factors contributing to low sperm counts, oligozoospermia, asthenozoospermia, and other semen abnormalities.

### 2.2. Stimulation and Transfer Protocol

Starting on Days 2 or 3 of the menstrual cycle, patients underwent controlled ovarian stimulation following the standard antagonist protocol [11,14] using recombinant FSH in combination with human menopausal gonadotropin (hMG).

The stimulation was monitored using transvaginal ultrasound and serum E2 levels. When the maximum follicle size reached 14–15 mm, ovulation was triggered with 0.25 mg of a gonadotropin-releasing hormone (GnRH) antagonist. Oocytes were retrieved 35–36 h after administration of the ovulation trigger. Immediately after receiving oocytes and spermatozoa, their morphological assessment was made. Mature oocytes with a pronounced first polar body (stage MII) were selected for fertilisation [15].

All oocytes were fertilised using the intracytoplasmic sperm injection (ICSI) procedure. The quality of resulting embryos was morphologically assessed 40–42 h (2 days), 72–74 h (3 days), and 120 h (5 days) after fertilisation.

Fresh embryo transfer was performed on Day 5 after fertilization. Prior to transfer, the quality of blastocysts was assessed by their size, using a scale from 1 to 5, by the state of the inner cell mass (ICM), using a scale from A to C, and by the size of trophoblast cells, using a scale from A to C [10,16,17]. Only embryos found to be euploid following PGT-A (see below) that were 3–5 mm in size with a multicellular ICM and a developed trophoblast were selected for transfer. Two embryos were transferred to women in each study group.

The advent of pregnancy was determined by assessing human chorionic gonadotrophin (hCG) in the blood on the 13–15th day after the embryo transfer and later with ultrasound when the foetal egg and the heartbeat of the embryo were detected.

### 2.3. PGT-A Procedure

The biopsy of blastocysts was performed on Day 3 after fertilisation at the stage of 6–10 blastomeres using Clinical Laser System Zilos-Tk (Hamilton Thorne, Beverly, MA, USA). Seven autosomes and both sex chromosomes were investigated for numeric abnormalities with FISH, using fluorescently labelled Vysis probes (Abbot Laboratories, Abbot Park, IL, USA): LSI 13 (Chromosome 13), CEP 15 (Chromosome 15), CEP 16 (Chromosome 16), CEP 17 (Chromosome 17), CEP 18 (Chromosome 18), LSI 21 (Chromosome 21), LSI 22 (Chromosome 22), CEP X (X chromosome), and CEP Y (Y chromosome).

Euploidy was defined as a complete set, haploidy as a single set, and polyploidy as three or more sets of chromosomes. Aneuploidy was defined as the presence of an abnormal number of chromosomes.

### 2.4. Statistical Analysis

The data obtained were processed using a Microsoft Excel spreadsheet from MS Office 365 package (Microsoft Corporation, Redmond, WA, USA). Pearson’s χ^2^ test and the Kolmogorov–Smirnov (K-S) test were used to determine significant differences between the study groups.

## 3. Results

The most common types of semen abnormality found in Group A and Group B were teratozoospermia and oligoastenoteratozoospermia, accounting for more than 50% of all pathologies detected (Table 2). There were no significant differences between the experimental groups with regard to each type of pathology observed.

As patients from Group B opted out of PGT-A testing, only embryos obtained for Group A and Group C underwent blastomere biopsy at Day 3 of cultivation. Overall, 880 embryos from Group A and 890 embryos from Group C were analysed for aneuploidies (Table 3).

The frequency of aneuploidy was found to be more than twice as high in patients with semen abnormality compared to the control group (13.6% vs. 5.8%, *p* < 0.001). However, no significant differences between and within the groups with regard to aneuploidies for particular chromosomes were detected. This could be due to several reasons, including heterogeneous causes of semen abnormalities and known limitations of the PGT-A detection technique employed in the study. It can also be that the biological defects underlying semen abnormalities do not selectively impact specific chromosomes but rather broadly affect cell chromosomal complement.

For all chromosomes studied, with the exception of Chromosome 21, both monosomies and trisomies were detected. Monosomy 21 was not observed in either group while Trisomy 21 was the most common type of aneuploidy in both Group A and Group C. Trisomy 15 and Monosomy 17 were not found in Group C. About 30% of aneuploid embryos in both groups contained complex aneuploidies—either multiple monosomies or trisomies or combinations of monosomies and trisomies.

The outcomes of the inclusion of PGT-A in the IVF treatment protocol of patients with abnormal semen parameters are presented in Table 4. Although there was no statistical difference in clinical pregnancy rate between the patients with or without abnormal semen parameters when PGT-A was included in the treatment protocol (47.3% vs. 56.2%), it was statistically significant when compared to Group B whose patients opted out of the preimplantation screening (28.2%). The latter was expected as PGT-A selection minimises the number of transferred aneuploid embryos, while in the control group, this proportion would be higher. Evidently, the potential reduction in embryo viability and implantation potential associated with the trophectoderm biopsy has a smaller effect on implantation and the clinical pregnancy rate than embryo aneuploidy.

The highest percentage of pregnancy loss was in patients from Group B. In this group, almost two out of three pregnancies ended in a miscarriage. In Group A and Group C, a significantly smaller number of clinical pregnancies ended up in miscarriages (25.0% and 11.9%, respectively).

In patients with abnormal semen parameters who did not consent to PGT-A testing, only 10% of IVF cycles resulted in LB. This parameter was more than three times lower than in similar patients who opted for PGT-A to be included in the treatment protocol. As expected, the highest birth rate observed was in control patients (Group C), where every second IVF cycle resulted in LB.

## 4. Discussion

The main finding of our work is that in couples where male partners had abnormal semen parameters, the inclusion of PGT-A into the treatment protocol had a marked positive effect on both the clinical pregnancy rate and the LB rate. Preimplantation genetic screening of embryos almost doubled the chances of pregnancy and tripled the LB rate in these patients.

Spermatogenesis is a combination of very complex processes of cell divisions involving intricate events like the duplication and recombination of genetic material. Errors at every stage can lead to sperm aneuploidy through various mechanisms, such as premature cell division, cell fusion, and chromosome breakage, to name a few. To make sure that sperm cells carry an accurate chromosome complement, most organisms evolved a specific control mechanism, the pachytene checkpoints, that monitor the fidelity of chromosome synapsis and the repair of DNA damage. This mechanism prevents meiotic nuclear division in cells that fail to complete meiotic recombination and chromosome synapsis, causing defective meiocytes to self-destruct, thus preventing the production of aneuploid gametes [18].

It has been long hypothesised that different types of human embryonic anomalies may be of meiotic or mitotic origin [19]. Meiotic abnormalities before fertilisation are the most likely mechanism of aneuploidy, which is universal for all embryonic cells. This may be due to the nondisjunction of entire chromosomes during meiosis I or II or the premature division of a chromosome into two sister chromatids during meiosis I, followed by their random separation. Mitotic disturbances can happen due to lack of divergence, endoreduplication, or anaphase delay, which most often occur during the first three divisions after fertilisation.

The effect of meiotic mechanism malfunctions on lower sperm counts in humans has been well documented [19]. An analysis of various stages of gametogenesis indicated that pachytene checkpoint I prevents the production of aneuploid gametes by inducing meiotic arrest of abnormal cells and, therefore, leads to oligospermia or azoospermia [4]. However, a small number of premeiotic abnormal cells can escape pachytene checkpoint I, reach meiosis, and produce chromosomally abnormal spermatozoa [20], which can be the cause of embryonic aneuploidy. The fact that, in our experiments, the percentage of aneuploid embryos in Group A was more than twice higher than in the control Group C (Table 3) lends support to this finding.

It has been known for some time that low sperm count and other types of sperm deficiency are directly linked to an increased rate of aneuploidy in sperm cells and, consequently, to embryonic aneuploidy. A significant negative correlation between sperm concentration and the estimated numerical chromosome aberrations in spermatozoa has been previously documented [21,22]. Saei and colleagues [23] observed, using FISH, that 55.46% of sperm in oligoasthenoteratozoospermic patients was aneuploid.

An increase in abnormal sperm FISH results in males with a decreasing sperm concentration and a positive effect of including PGT-A in the IVF protocol for treating such patients has been recently shown [24]. Couples with abnormal sperm FISH results for the male exhibited better clinical outcomes after PGT-A, suggesting a potential contribution of sperm to embryo aneuploidy. Our data on the effect of PGT-A on the clinical pregnancy rate and LBs in couples where male partners had abnormal semen parameters also agree with these findings. In couples from Group B who opted out of PGT-A screening, almost two-thirds of clinical pregnancies resulted in pregnancy loss. In patients from Group A and Group C (control patients), pregnancy loss was significantly lower (Table 3).

Several other studies have suggested that spermatozoa aneuploidy can be one of the reasons for repeated abortion and recurrent implantation failure [25,26]. These incidences are especially high in patients with oligoasthenoteratozoospermia. In another study, Zidi-Jrah and colleagues [27] showed that sperm with abnormal morphology was significantly higher in patients with recurrent pregnancy loss.

In our study, the inclusion of PGT-A into the IVF treatment protocol had a distinct effect on embryo progress to delivery in couples where the male partner had semen abnormalities. Following fresh embryo transfer, couples in Group A had a significantly higher clinical pregnancy rate than those from Group B (47.3% vs. 28.2% *p* < 0.01). This parameter was somewhat higher in patients from the control, Group C, but no significant differences in clinical pregnancy rate between patients with and without semen abnormalities were observed when euploid embryos were selected for transfer using PGT-A.

Still, there was a marked difference in the number of miscarriages between these groups. Every fourth confirmed pregnancy in Group A resulted in a loss, while in Group C, the figure was only half of that. The significantly higher pregnancy loss in Group A compared to the control may indicate that even when euploid embryos are transferred into the womb, there may exist some other causes associated with abnormal semen parameters, which play a critical role in the inability of pregnancy to progress to LB.

There was an especially pronounced positive effect of embryonic screening on LB. In more than a third of instances, IVF treatment augmented with PGT-A resulted in the birth of a healthy child in couples where the male partner had semen abnormalities. However, only 10% of IVF cycles in these patients resulted in LB if PGT-A was not used. Patients in the control group had the highest LB rate, where almost every second cycle resulted in LB.

Our study confirms previous observations that PGT-A should be used not only for screening and diagnostics to achieve pregnancy and decrease the chances of miscarriage but also as an efficient treatment tool for IVF patients [28]. This is especially true for couples where the male partner suffers from sperm disorders. For these patients, including PGT-A screening within the framework of ART allows selecting embryos with no numerical chromosomal abnormalities for transfer, increasing the chances of successful implantation, reducing the risk of pregnancy loss, and achieving the birth of a healthy child.

Although our study showed the positive outcome of PGT-A in treating patients with sperm abnormalities, some other reports showed no such effect [29,30]. In a recent study [29], the euploidy rate of the obtained blastocysts and live birth were found to be independent of sperm quality. In contrast, in our study there was a significant difference between the type of sperm abnormality and these parameters. There may be several reasons for such a discrepancy, attributed to differences in selecting patient populations, methodology, specific outcome measures, and some other factors. Nevertheless, the issue of the effect of PGT-A on outcomes of IVF in patients with male factor infertility needs to be addressed by further studies.

PGT-A is known to carry certain risks, including potential embryo damage during biopsy, which may affect the implantation and clinical pregnancy rates. However, despite these drawbacks, our data indicate the importance of PGT-A for successfully treating patients with male factor infertility. The results obtained in the current study add further evidence that couples where male partners have a pathologically low sperm count and other semen abnormalities have significantly higher rates of embryonic aneuploidy and, consequently, elevated risks of miscarriage and chromosomal pathology in the newborn. Proper, non-directive, and unbiased genetic counselling should be offered to such couples, emphasising the increased risk of chromosome aneuploidy in their offspring and the importance of PGT-A in preventing this potential risk. The couples should be provided with the necessary information about the reproductive options open to them, including available genetic testing methodologies and procedures.

## 5. Conclusions

Advances in ART and especially the development of ICSI have revolutionised the fertility treatment of men with severely impaired sperm parameters. By reducing requirements for sperm number, quality, and motility, they increase the chances of biological fatherhood in such patients [14]. The inclusion of PGT-A in the treatment protocol for patients with semen abnormalities significantly improves the chances of clinical pregnancy and live birth. PGT-A allows the selection of only chromosomally normal embryos for transfer, thus increasing the positive outcome of fertility treatment even when suboptimal sperm has to be used for fertilisation.

## Figures and Tables

**Table 1 biomedicines-13-01191-t001:** Demographics and outcome data on PGT-A embryo assessment in all study groups.

Parameter	Group A		Group B		Group C	
No. of couples	110		110		105	
	♀	♂	♀	♂	♀	♂
Average age, years (mean ± SD)	30.4 ± 0.5	34.1 ± 0.5	31.1 ± 0.6	35.2 ± 0.6	32.0 ± 0.6	34.2 ± 0.6
Minimum age, years	21	25	21	27	21	22
Maximum age, years	43	50	43	52	43	50
Distribution of the number of participants of the study by age group						
	Group A		Group B		Group C	
Age group, years	♀	♂	♀	♂	♀	♂
20–24	19	-	17	-	18	1
25–29	35	28	33	27	23	29
30–34	28	28	30	30	24	20
35–39	20	33	17	21	22	31
40–44	8	13	13	19	18	15
45–49	-	7	-	8	-	8
50–55	-	1	-	5	-	1
Outcome of PGT-A embryo assessment						
	Group A		Group B		Group C	
No. of embryos subjected to PGT-A	880		-		890	
No. of euploid embryos identified by PGT-A	760		-		838	
No. of embryos transferred	220		220		210	

**Table 2 biomedicines-13-01191-t002:** Types of semen abnormality observed in the experimental groups.

Type	Group A (n = 110)		Group B (n = 110)		*p*
	Patients Observed	% of Total	Patients Observed	% of Total	
Oligozoospermia	11	10.0%	9	8.2%	0.639
Asthenozoospermia	14	12.7%	12	10.9%	0.676
Oligoastenozoospermia	16	14.5%	21	19.1%	0.367
Teratozoospermia	32	29.1%	29	26.4%	0.651
Oligoastenoteratozoospermia	37	33.6%	39	35.5%	0.777

**Table 3 biomedicines-13-01191-t003:** Results of PGT-A testing of embryos in patients with (Group A) and without (Group C) abnormal semen parameters.

Type of Aneuploidy	Group A (n = 880)		Group C (n = 890)		*p*
	Number Observed	% of Total per Group	Number Observed	% of Total per Group	
Monosomy 13	3	0.3%	3	0.3%	0.693
Trisomy 13	6	0.7%	5	0.6%	0.748
Monosomy 15	4	0.5%	1	0.1%	0.364
Trisomy 15	3	0.3%	-	-	-
Monosomy 16	4	0.5%	1	0.1%	0.364
Trisomy 16	6	0.7%	2	0.2%	0.281
Monosomy 17	3	0.3%	-	-	-
Trisomy 17	3	0.3%	1	0.1%	0.609
Monosomy 18	5	0.6%	1	0.1%	0.215
Trisomy 18	8	0.9%	3	0.3%	0.219
Trisomy 21	12	1.4%	7	0.8%	0.239
Monosomy 22	7	0.8%	3	0.3%	0.332
Trisomy 22	8	0.9%	3	0.3%	0.219
Monosomy X	7	0.8%	3	0.3%	0.332
XXY	5	0.6%	1	0.1%	0.215
XYY	4	0.5%	2	0.2%	0.672
Multiple Monosomies	11	1.3%	6	0.7%	0.214
Multiple Trisomies	9	1.0%	4	0.4%	0.158
Complex Aneuploidies	12	1.4%	6	0.7%	0.148
Total	120	13.6%	52	5.8%	<0.001

**Table 4 biomedicines-13-01191-t004:** The effect of PGT-A testing on IVF treatment outcomes in patients with and without abnormal semen parameters.

IVF Outcome	Group A n = 110		Group B n = 110		Group C n = 105		*p*		
	Number Observed	%	Number Observed	%	Number Observed	%	Group A vs. Group B	Group A vs. Group C	Group B vs. Group C
Clinical pregnancy per cycle	52	47.3%	31	28.2%	59	56.2%	0.008	0.191	<0.001
Pregnancy did not occur	58	52.7%	79	71.8%	46	43.8%	0.001	0.191	<0.001
Live birth per cycle	39	35.5%	11	10.0%	52	49.5%	<0.001	0.038	<0.001
Pregnancy loss (all causes)	13	25.0% ^†^ 11.8% ^‡^	20	64.5% ^†^ 18.2% ^‡^	7	11.9% ^†^ 6.7% ^‡^	0.05 0.145	0.05 0.199	<0.005 0.014

^†^—from the number of clinical pregnancies; ^‡^—from the number of couples in the group.

## Data Availability

All data generated and analysed during this study are included in this published article.

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
