# Peer review of "Preimplantation Testing of Human Blastomeres for Aneuploidy Increases IVF Success in Couples Where Male Partners Had Abnormal Semen Parameters"

_biomedicines, 2025, doi:10.3390/biomedicines13051191_

Round 1
Reviewer 1 Report
Comments and Suggestions for Authors
The authors evaluate the benefits of preimplantation embryo tests for aneuploidies (PGT-A) in IVF procedures performed due to male infertility. They demonstrate that FISH-based PGT-A performed on patients with abnormal sperm parameters shows an elevated rate of aneuploid embryos compared to the control group with normal spermograms. Live birth rates were considerably higher in patients with abnormal spermograms who underwent PGT-A, reaching 35% compared to 10% without the test. For patients with normal spermograms, the live birth rate was approximately 50%.
Overall, the study is well documented and reported, but some issues need to be clarified before the manuscript can be accepted.
- A number of studies report an absence of any benefits of the PGT-A for the patients with male factor infertility: Wang et al., 2025 https://doi.org/10.1016/j.fertnstert.2025.02.014, or Mazzilli et, 2017: https://doi.org/10.1016/j.fertnstert.2017.08.033 , and others. The authors need to review these finding and explain the discrepancy.
- The authors mention in the discussion that according to the literature, in patients with oligoasthenoteratozoospermia (i.e. with low sperm count, low motility, and abnormal morphology), 55% of sperm is aneuploid (line 240). What are aneuploidy rates for other types of sperm abnormalities mentioned in the study (oligozoospermia, asthenozoospermia, teratozoospermia)?
- It is of interest to evaluate the total aneuploidy rates and live birth rates observed after PGT-A separately for each type of semen abnormality.
Author Response
We are grateful to Reviewer 1 for valuable comments and suggestions to improve our manuscript. Below please find our response to the comments.
Comment 1: A number of studies report an absence of any benefits of the PGT-A for the patients with male factor infertility: Wang et al., 2025 https://doi.org/10.1016/j.fertnstert.2025.02.014, or Mazzilli et, 2017: https://doi.org/10.1016/j.fertnstert.2017.08.033 , and others. The authors need to review these finding and explain the discrepancy.
Reply
A paragraph dealing with the discrepancy between our study and the studies mentioned by the Reviewer was added to the manuscript.
Comment 2: The authors mention in the discussion that according to the literature, in patients with oligoasthenoteratozoospermia (i.e. with low sperm count, low motility, and abnormal morphology), 55% of sperm is aneuploid (line 240). What are aneuploidy rates for other types of sperm abnormalities mentioned in the study (oligozoospermia, asthenozoospermia, teratozoospermia)?
Reply
There are several studies which describe aneuploidy rates for the other types of sperm abnormalities. Young et al (2022) [doi: 10.3389/fendo.2022.1072176.] estimated the aneuploidy rate of blastocyst in teratozoospermic and asthenozoospermic patients as 55.0% and 46.7% respectively. The effect of oligozoospermia on aneuploidy is similar (e.g. Rodrigo et al. 2019 [doi.org/10.1093/biolre/ioz125]).
Comment 3: It is of interest to evaluate the total aneuploidy rates and live birth rates observed after PGT-A separately for each type of semen abnormality.
Reply
This is an interesting and important question. Although it was not the purpose of the current study, and a bigger sample size is needed to evaluate such an effect we are currently working on collecting more data. Hopefully, we will be in a position to assess the effect of the semen abnormality type on the live birth and total aneuploidy rates sometime next year, when a sufficient amount of data is collected.
Reviewer 2 Report
Comments and Suggestions for Authors
The authors discuss the IVF success rates in couples where male partners have
abnormal semen parameters after preimplantation testing of blastomeres for aneuploidy.
Comment 1
Lines 107-109: "Couples in which one of the spouses smoked or had alcohol or drug addiction were also excluded from the study group"
Please comment on ths statement as far as the exclusion of the male partner. Smoking, alcohol consumption etc. are parameters that affect sperm qulaity, thus could result in oligospermia, asthenospermia and so on.
Comment 2
Please disscuss or analyze further the following findings.
Lines 173-175: "The frequency of aneuploidy was found to be more than twice as high in patients with semen abnormality as compared to the control group (13.6% v 5.8%, p<0.001). However, no significant differences between and within the groups with regard to aneuploidies for particular chromosomes were detected"
Lines 184-188: Although there was no statistical difference in clinical pregnancy rate between the patients with or without abnormal semen parameters when PGT-A was included in the treatment protocol (47.3% v 56.2%), it was statistically significant when compared to Group B whose patients opted out of the preimplantation screening (28.2%).
Comment 3
Please refer to PGT-A dangers generally and if you had any in your experimental design.
Author Response
We are grateful to Reviewer for valuable comments and suggestions how to improve our manuscript. Below please find our replies to the comments.
Comment 1
Lines 107-109: "Couples in which one of the spouses smoked or had alcohol or drug addiction were also excluded from the study group"
Please comment on ths statement as far as the exclusion of the male partner. Smoking, alcohol consumption etc. are parameters that affect sperm qulaity, thus could result in oligospermia, asthenospermia and so on.
Reply:
A comment about excluding the male partners who smoked or had an addiction was provided.
Comment 2
Please disscuss or analyze further the following findings.
Lines 173-175: "The frequency of aneuploidy was found to be more than twice as high in patients with semen abnormality as compared to the control group (13.6% v 5.8%, p<0.001). However, no significant differences between and within the groups with regard to aneuploidies for particular chromosomes were detected"
Reply
The text was amended to include a brief discussion of this paragraph.
Lines 184-188: Although there was no statistical difference in clinical pregnancy rate between the patients with or without abnormal semen parameters when PGT-A was included in the treatment protocol (47.3% v 56.2%), it was statistically significant when compared to Group B whose patients opted out of the preimplantation screening (28.2%).
Reply
The text was amended to include a brief discussion of this paragraph.
Comment 3
Please refer to PGT-A dangers generally and if you had any in your experimental design.
Reply
Information on the main danger of PGT-A to the embryo (biopsy-associated damage) was added to the text. As this is an intrinsic feature of PGT-A we took it into account as it may cause some reduction of the implantation and clinical pregnancy rates. Our data indicate that even with a possible effect on these parameters, PGT-A was beneficial for patients with semen abnormalities.